# Interaction Modes of Microsomal Cytochrome P450s with Its Reductase and the Role of Substrate Binding

**DOI:** 10.3390/ijms21186669

**Published:** 2020-09-11

**Authors:** Francisco Esteves, Philippe Urban, José Rueff, Gilles Truan, Michel Kranendonk

**Affiliations:** 1Center for Toxicogenomics and Human Health (ToxOmics), Genetics, Oncology and Human Toxicology, NOVA Medical School/Faculty of Medical Sciences, Universidade NOVA de Lisboa, 1169-056 Lisbon, Portugal; jose.rueff@nms.unl.pt; 2TBI, Université de Toulouse, CNRS, INRAE, INSA, CEDEX 04, 31077 Toulouse, France; philippe.urban@insa-toulouse.fr (P.U.); gilles.truan@insa-toulouse.fr (G.T.)

**Keywords:** NADPH-cytochrome P450 reductase, microsomal cytochrome P450, CPR-FMN-domain, protein–protein interaction, CYP-substrate, electron transfer

## Abstract

The activity of microsomal cytochromes P450 (CYP) is strictly dependent on the supply of electrons provided by NADPH cytochrome P450 oxidoreductase (CPR). The variant nature of the isoform-specific proximal interface of microsomal CYPs implies that the interacting interface between the two proteins is degenerated. Recently, we demonstrated that specific CPR mutations in the FMN-domain (FD) may induce a gain in activity for a specific CYP isoform. In the current report, we confirm the CYP isoform dependence of CPR’s degenerated binding by demonstrating that the effect of four of the formerly studied FD mutants are indeed exclusive of a specific CYP isoform, as verified by cytochrome *c* inhibition studies. Moreover, the nature of CYP’s substrate seems to have a modulating role in the CPR:CYP interaction. In silico molecular dynamics simulations of the FD evidence that mutations induces very subtle structural alterations, influencing the characteristics of residues formerly implicated in the CPR:CYP interaction or in positioning of the FMN moiety. CPR seems therefore to be able to form effective interaction complexes with its structural diverse partners via a combination of specific structural features of the FD, which are functional in a CYP isoform dependent manner, and dependent on the substrate bound.

## 1. Introduction

Human mitochondrial and microsomal cytochromes P450 (CYP) form a superfamily of membrane-bound hemeproteins with monooxygenase activity [1,2]. Microsomal CYP isoforms are dependent on NADPH cytochrome P450 oxidoreductase (CPR)—containing flavin adenine dinucleotide (FAD) (reductase) and flavin mononucleotide (FMN) (transporter) domains—for supply of electrons in their catalytic activities to metabolize a myriad of compounds, both endobiotics including vitamins, steroids and hormones, and xenobiotics, such as therapeutic drugs, carcinogens and environmental chemicals [3,4,5].

The study of affinity and binding in the CPR:CYP microsomal monooxygenase system is crucial for a better understanding of the structural features which guide the formation of an efficient enzyme complex, determinant for electron transfer (ET) from CPR to CYP, thus potentially modulating physiologically important microsomal CYPs activities. Although all CYPs are likely to diverge from a common ancestor gene, their interactions with redox partners seem to have evolved differently, depending on the cellular localization and type of substrates to metabolize [2,6]. Mitochondrial CYPs exhibit conserved basic residues at their proximal side for effective interaction with their cognate redox partner: [2Fe-2S] adrenodoxin (Adx) [7]. Contrarily, microsomal CYPs do not possess such signature sequences to guide their interaction with the FMN-domain (FD) of CPR for ET [6]. This suggests that CPR:CYP electron transport system is more versatile and does not depend on uniquely conserved structural features present on CPR–CYPs interaction interface. Instead, it relies on a degenerate interaction between the FD and the proximal side of the structurally diverse microsomal CYP isoforms [6,8,9,10].

The precise mechanism governing this degenerated protein-protein interaction and how CPR can sustain efficient ET with so many structurally diverse partners has been the subject of many studies over the last decades (reviewed in [2,5,10]). The overall steady-state ET from CPR to microsomal CYPs seems to occur via a concerted mechanism involving the FD and FAD-domain within CPR (inter-flavin ET) and the FD and CYP proximal side (inter-protein ET) [3,4,5]. Multiple studies reported by our group and others indicate that effective docking and ET from CPR to its many redox partners are enabled by: (i) The existence of multiple open conformations, the result of CPR’s open-closed dynamics, that can be sampled by its diverse redox partners [11,12,13,14,15]; and (ii) affinity probing of these open conformers, guided by electron receptor specific affinity for the degenerated interface/binding site of CPR [6,8,9,16,17]. Together with the membrane anchoring segment, implicated in the preliminary association of the CPR with CYPs and in substrate entrance/binding to CYP’s substrate-access channel [18,19,20,21], these features are thus required for an effective ET by CPR to multiple CYPs.

However, these “one-serve-all” types of interactions may still be dependent on the CYP isoform. It is likely that multiple CPR conformations and binding modes are probed in a selection process by which each of the structurally diverse CYP isoforms binds to (a) more favorable conformation(s) of the CPR [5]. Additionally, the in vivo existence of a vast excess of CYPs over CPR, implies competition between individual CYP isoforms in binding to CPR. This evidences the role of affinity in the formation of effective transient enzyme complexes, rather than stable complexes [11,22,23].

Through structural and functional studies, specific patches of acidic residues surrounding the flavin isoalloxazine ring and on the binding interface of the FD were identified. These acidic residues drive the effective docking between CPR and its multiple electron acceptors via electrostatic interactions (Appendix A) [14,17,24,25,26]. However, the relative importance of these residues for interaction seems to be dependent on the CPR:redox-partner couple [2,5]. Additionally, basic residues on the proximal side of CYPs seem to steer the negatively charged convex binding-interface of the FD to CYP’s concave docking site near the heme moiety [9]. Interestingly, structural information on CYP substrate/ligand complexes and conformational dynamics for substrate access demonstrated that CYP’s active site adapts to substrates of diverse sizes and shapes, which in turn may have implications in the CPR:CYP interaction as the structural rearrangement associated with substrate binding was shown to extends its effects to the proximal side of CYPs [1,2,27,28,29,30,31].

We previously studied the underlying molecular mechanisms on CPR:CYP interaction and ET, by creating and selecting seven mutants in the FD of human CPR, each supporting a gain in activity for a specific CYP isoform [16]. Most of the identified positions co-localize with specific acidic stretches—formerly indicated to be involved in CPR:CYP interactions—and/or with mutations found in naturally occurring CPR variants—previously shown to cause CYP isoform dependent effects [4,22,32,33,34,35]. The present study aims: (i) To explore CPR’s degenerated binding mode by demonstrating that the generated FD mutants of CPR are exclusive of a specific CYP isoform and thus is detrimental to interaction of CPR with other CYP forms; (ii) to determine the role of substrate-binding in CYPs in modulating this specific recognition. From the set of previously identified CYP-specific CPR mutants [16], we selected P117H and G144C (specific for CYP1A2), G175D (specific for CYP2A6) and N151D (specific for CYP3A4) and crossed-combined each of them with the two other CYPs, i.e., the ones not used in their selection. Effects were evaluated using extended enzyme kinetic analysis with different fluorogenic standard probes, cytochrome *c* (cyt *c*) competition, and multiple structural (in silico) analyses. The results presented herein further confirm the preferential interaction of CYP isoforms with FD mutants and describe the structural features that promote the CPR–CYP recognition in an isoform specific manner, thus potentially modulating CPR’s promiscuous interactions with structurally diverse CYP isoforms and CYP-substrate binding states.

## 2. Results

### 2.1. Characterization of the Membrane Fractions

The characterization of contents of heterologous expressed proteins found in the membrane fractions (isolated using the bacterial CPR/CYP co-expression system, BTC) [22,36] containing the wildtype CPR (CPR_wt_) and mutant CPRs (CPR_mut_) combined with CYP1A2, 2A6 or 3A4 is shown in Table 1. Co-expression of CPR_mut_ with CYP1A2, 2A6, or 3A4 was achieved with a stoichiometry approximating the one obtained with CPR_wt_ for each CYP, demonstrating no significant differences (*p* > 0.05). As such, any difference detected in the CYP activities can be ascribed to differences in the CPR_mut_:CYP coupling. Indeed, the CPR:CYP stoichiometries measured in membrane fractions were similar to those described previously for the BTC system in our former studies [11,16,33,36,37,38,39], and comparable with the ranges of those reported for human liver microsomes, which range from 1:2 to 1:13 [40,41]. These ratios evidence a submolar presence of CPR vs. CYP, which is recapitulated in the BTC system, as already demonstrated in our former studies [11,22,36,39].

### 2.2. Detailed Enzyme Activity

#### 2.2.1. Reaction Velocities of the Different CPR_mut_:CYP Combinations

Comparison of the kinetic parameters for each CPR_mut_/CYP couple are presented in Figure 1 and detailed numbers in Appendix A. We confirm the significantly higher relative turnovers of these mutants in combination with specific CYPs, namely CPR_P117H_:CYP1A2 and CPR_G144C_:CYP1A2 [ethoxyresorufin (EthR) O-deethylation, EROD], CPR_G175D_:CYP2A6 (coumarin 7-hydroxylation, C7H) and CPR_N151D_:CYP3A4 [dibenzylfluorescein (DBF) O-debenzylation, DBODF], as was observed in our previous study [16]. Except for CPR_G175D_:CYP3A4 (increased *k*_cat_, 1.22 X, *p* < 0.005) (Figure 1C), all the new CPR_mut_:CYP combinations demonstrated equal (CPR_P117H_:CYP2A6; CPR_N151D_:CYP2A6) or significantly lower activities (CPR_G175D_:CYP1A2; CPR_N151D_:CYP1A2; CPR_G144C_:CYP2A6; CPR_P117H_:CYP3A4; CPR_G144C_:CYP3A4) when compared to the corresponding CPR_wt_:CYP couples. According to this data, one single mutation in the FD is able to induce increased ET efficiency for one specific CPR:CYP combination, while maintaining or decreasing this activity for other couples. In few cases, the apparent affinity constant (K_M_) changed with the FD mutant CPRs. The apparent affinity constant of the CYP enzyme complex is a combination of multiple factors such as substrate entrance, binding and product egress but also ET rates. We thus hypothesize that mutations in the FD, by either affecting the binding of CPR to CYPs or directly modifying ET, may affect the apparent affinity constants.

#### 2.2.2. Reaction Velocities of CPR_mut_:CYP1A2 Using Different Substrates

Microsomal CYP turn-overs can be modulated by many parameters and, amongst those, the efficiency of the complex formation between CPR and CYP and the substrate nature and presence in the active site are known to be critical [5,42]. We thus compared the effects of the four FD mutations of CPR with CYP1A2 using two other substrates (Figure 2 and Appendix A). Globally, the effects of the mutations are substrate dependent: The relative *k_cat_* profiles observed for the CPR_mut_:CYP1A2 in [3-cyano-7-ethoxycoumarin (CEC) O-dealkylation, CECOD] (Figure 2A) and DBODF (Figure 2B) are distinct from the ones of EROD. Contrarily to the activity profiles observed for EROD (Figure 1A, i.e., increased CYP1A2 activity with P117H and G144C; *p* < 0.005), the *k*_cat_ of CYP1A2 generated with these CPR mutants demonstrated CPR_wt_-like profiles for CECOD (Figure 2A) and DBODF (Figure 2B). The G175D mutant (selected previously for increased CYP2A6-C7H activity) showed a decrease in CYP1A2-EROD (*p* < 0.05) and -CECOD (*p* < 0.05) activities, when compared with the CPR_wt_. Inversely, this mutant increased considerably the CYP1A2-DBODF activity (*p* < 0.005). A decrease in CYP1A2-EROD (*p* < 0.005) and -CECOD (*p* < 0.005) activities and a CPR_wt_-like CYP1A2-DBODF activity were observed for the N151D mutant (selected previously for increased CYP3A4-DBODF activity).

When comparing the relative *k*_cat_ profiles of the different reactions, DBODF, a typical CYP3A4 substrate, showed turnover profiles (Figure 2B) distinct from the other substrates studied. This may be explained by DBF’s less favorable chemical structure, when compared to the other, more typical CYP1A2 substrates studied. P117H, G144C and N151D demonstrated relative DBODF turnovers very similar to the ones of CPR_wt_:CYP1A2, while for G175D we observed a completely different pattern. For this specific mutant, relative *k*_cat_ values increased dramatically (1.83 X, *p* < 0.005). This was not observed, either for the other two substrates or by any of the other CPR_mut_:CYP1A2 combinations. Additionally, the CPR_mut_:CYP1A2 reaction velocities profiles for the CYP1A2 mediated de-alkylation of methoxyresorufine (MROD) (Appendix A) were identical to the ones observed for the EROD reaction. These results are not surprising because substrates (typical CYP1A2 substrates) are similar and deviate only by one CH_2_ group.

### 2.3. Cytochrome c Competition with CPR_mut_/CYP

In two recent studies, we applied cytochrome *b*_5_ (CYB5) to probe affinity differences between CPR mutants and CYP [11,16]. Although informative, the inhibition data were difficult to interpret due to a dual effect by CYB5, namely an initial stimulating effect on CYP activity (depending on the CYP-isoform) and subsequent inhibitory effects at higher CYB5 levels when CPR binding is prevented [16]. An alternative competitive inhibitor is cyt *c*. Structural and functional studies of the FD binding site of rat CPR evidenced that the surface patches involved in protein:protein contacts with both CYP and cyt *c* are probably overlapping (detailed in Appendix A) [5,14,17,24,25,43,44,45,46,47,48]. Therefore, cyt *c* competition experiments may shed light on differences in affinity between the various FD mutants when binding to different CYPs, without any confounding stimulatory effects when CYB5 is applied for this purpose. Relative velocities (*k_obs_* mutant/*k*_obs_ wild-type) were determined for the four FD mutants with various cyt *c*/CPR ratios for CYP1A2, 2A6, or 3A4-mediated reactions (Figure 3). Globally, cyt *c* seems to have an inhibitory effect on CYP dependent reactions, suggesting direct competition between cyt *c* and CYP for CPR binding. The relative velocity traces of the CPR_mut_:CYP couples CPR_P117H_:CYP1A2, CPR_G144C_:CYP1A2, CPR_G175D_:CYP2A6 and CPR_N151D_:CYP3A4 show less inhibited activities than those from the corresponding CPR_wt_/CYP couples. Except for CPR_N151D_:CYP2A6 (Figure 3B), which showed a pronounced inhibition by cyt *c*, the profiles of all other CPR_mut_:CYP couples were comparable to the ones observed for the CPR_wt_:CYP. One-way analysis of variance demonstrated significant differences (*p* < 0.05) between the distribution profile of the means of the activity traces of CPR:CYP1A2 (Figure 3A) and CPR:CYP2A6 (Figure 3B) couples.

These results confirm that the competition exerted by cyt *c* with CYP for CPR-binding (i.e., ability to inhibit CPR:CYP interaction and ET) depends on the strength of interaction between CPR and a particular CYP, thus reinforcing the notion that the four mutants have indeed increased affinities for the CYP with which they were selected. It is also most probable that the level of ionic interactions in the CPR:CYP complex are CYP isoform dependent, which our former study demonstrated [11,16] and current data confirmed.

### 2.4. Structural Analysis of CPR Mutants

In order to interpret the biochemical properties of the mutants with structural modifications, we performed molecular dynamics simulations. Four independent simulations were run for the wild-type and each of the mutants, in water, during 80 ns. A first analysis of root mean square deviations (RMSD) along the progress of the simulations revealed a common pattern: A rise in the RMSD in the first 20 ns, corresponding to an increase in the energy as the starting coordinates were directly issued from a crystal structure and then a stabilization of the RMSD corresponding to a stable energy state, with rather small variations between the different simulations (Appendix A). Comparisons of the average structures from the simulations with the FMN domain from the crystal structure reveals minor deviations which mostly affect helix A of the FMN domain. This helix, located at the N-terminus, is probably more subjected to translation movements compared to the rest of the structure. In one of the simulations (P117H, simulation 1), the time to reach the energy equilibrium is longer (40 ns instead of ~20 ns). In the average structure from this simulation, the D helix is slightly tilted toward the solvent. The average RMSD value for N151D in simulation 3 is also slightly higher, which might reflect a larger deviation from the starting structure. However, for both forms and simulations, the distribution of RMSD after the equilibrium is similar to the others, implying that the equilibrium, when reached, was the same for both particular simulations. In any case, the strongest deviations in RMSDs do not cluster (albeit possibly for P117H simulation) close to the mutated positions (Appendix A). Therefore, the various selected mutations do not, in an 80 ns simulation, perturb the overall structure or folding of the FD, but probably have rather specific and localized effects. We thus looked more closely into the different simulations in order to reveal these localized effects and link them to known properties for interactions between CPR and CYPs.

We first examined the effects of P117H and N151D mutations (Figure 4) on the pKa of acidic residues around both positions. For P117H, the presence of a histidine at the position 117 leads to a drastic change of the pKa of D116 and E118, but not E119 (Figure 4A). This change can be explained by the short distances between the ND1 (*tele* or τ in IUPAC numbering) atom of H117 and the oxygen atoms of the carboxylic functions of either D116 in simulation 1 and 4 (Appendix A) or E118 in simulation 2, 3, and 4 (Appendix A), leading to potential hydrogen bonds in both simulations (Appendix A). This difference between the two simulations is interesting as it reveals the two possible states of the sidechain of H117 which may be interchanged in longer simulations times. In any case, these changes propagate to the electrostatic potential which is less negatively charged in the H117 simulations than the one present in the wild-type simulations (Figure 4B). On the contrary, the presence of an aspartate residue instead of an asparagine at position 151 does not seem to modify the pKa of the surrounding acidic residues (Figure 4C). However, the pKa of D151, while being one pH unit higher than the theoretical value of aspartate (Figure 4C), introduces a supplementary negative charge in this area (Appendix A). This is also reflected on the electrostatic potential (Figure 4D), where, contrarily to P117H mutant (Figure 4B), the added negative charge leads to a quite visible increase of the negative potential in the same region. Overall, both mutations affect the electrostatic potential of the exact same area, although in a complete opposite way, decreasing the charge density for P117H and increasing the negative potential for N151D. Interestingly, D116 and E118 were formerly identified to be involved in hydrogen bonds with R440 in a docking model of the human FD of CPR with CYP2D6 [49,50]. Furthermore, a mutation of this arginine residue into histidine leads to a 5- to 100-fold decrease of CYP2D6 activity [49].

For G114C mutation, the situation is quite different as the glycine residue at this position in CPR is well conserved amongst species. As seen in Figure 5A, the cysteine at this position extends toward the surface but does not seem to perturb the positions of the residues surrounding the FMN moiety except for E145 for which the sidechain moves backward to the FMN. Analysis of the pKa of the ionizable residues close to the FMN (E145, E182 and H183) reveals no significant modifications compared to the wild-type (Appendix A). A more detailed analysis of the positioning of the FMN in this pocket (Appendix A) shows slight modifications of certain distances: Longer average distances between the OH atom of Y143 or the N atom of C144 (vs. G144) and the C7M or C8M atoms of FMN in the G144C mutant compared to the wild-type, longer distances between the CG2 of V191 and C7M, C8M, N3, and N5 atoms of FMN. We can thus conclude that the mutation G144C modifies slightly the positioning of the FMN moiety, which, in return, may impact the redox potential and electron transfer properties of the cofactor. However, the most drastic change can be seen on the electrostatic potential of the surface around the FMN (Figure 5B) which is much less negative than in the wild-type, due to probably both the movement of the E145 sidechain and the presence of the sulfur atom of C144 around. Such a change can also affect the redox potential of the FMN cofactor but also the positioning of the whole FD within the complex with CYPs.

Quite surprisingly, for the G175D mutant (Figure 5C,D), the addition of an acidic residue on β-strand 3 [45] does not modify drastically the structure of this region. The D175 residue is fully embedded in the structure and its modelled pKa value (~9.0) is more than 5 pH units higher than the theoretical value for aspartate residues (3.8), a quite logical value considering the total absence of water molecules surrounding this aspartate. Consequently, the presence of an aspartate does not modify the electrostatic potential around the mutated position. However, the OD1 atom of D175 is at ~3.1 Å of the N atom of T142 strongly suggesting that a hydrogen bond between β-strand 4 and α-helix E may be created. Analysis of the various φ and ψ angles for β-strands 3 and 4 shows little effects of the mutation on the overall secondary structures around the mutation position (Appendix A). The average φ angle value of D175 is smaller (Appendix A), demonstrating a small but visible distortion of the β-strand 3 (Figure 5D, green for WT, magenta for G175D). This is also reflected in the average φ values for A141 and M140 (Appendix A) where the changes of the average φ angle value are a bit smaller or bigger (respectively) than for D175. Furthermore, for A141, the deviations around the average φ values are bigger than for the wild-type, suggesting more flexibility at the end of the β-strand 4 (Appendix A). These changes impacted the residues surrounding the FMN moiety: The average distances from the OH atom from Y143 and N atom of G144 to the C7M and C8M atoms of FMN were increased (Appendix A). Overall, the effects introduced by 3 of the analysed mutations (P117H, G144C, N151D) lead to visible electrostatic surface changes, while, for the G175D mutation, no changes in the electrostatic could be detected but possible changes in the mobility of the residues around the FMN moiety are likely to occur.

## 3. Discussion

ET between CPR and its partners are mediated by weak and transient complexes that are highly dynamic [5,42] and can be described as an ensemble of various protein orientations rapidly interchanging between themselves [44]. Formation of these complexes is mainly guided by long-range electrostatic interactions between complementary charges on the interacting surfaces [5,14,24,25,31,43,45,46].

Several studies questioned the role of specific residues of the FD of CPR in the complex formation, either using site-specific modifications [3,14,17] or from analysis of naturally occurring variants of CPR (reviewed in [4,32]). In a previous study [16], we demonstrated that selected FD mutations can increase individual CYP isoforms activities. In the present study, we questioned the specificity of four (P117H, G144C, N151D, and G175D) of the previously generated mutations toward CYPs by analyzing the effect of each individual mutation on the three other CYPs and using different substrates. As expected, the majority of those mutations had either no or detrimental effects when tested with other CYP isoforms and can thus be considered CYP specific. It is known that CPR can interact with distinct CYPs and promote ET to several isoforms simultaneously. Our data underpin the notion that these “degenerated” interactions can be modulated by specific positions of the FD, increasing the interactions of CPR with a specific CYP while, at the same time, decreasing the formation of interacting complexes with other CYPs.

However, surprisingly, in the case of the CYP1A2, mutations that had negative effects with EROD and CECOD had neutral (N151D) or enhancing effects (G175D) when DBODF was used as substrate. This implies that the structural determinants present in the proximal side of CYPs can be modulated by the type of substrate, which, in return, may change CPR:CYP interactions, potentially annihilating the specific effects of the FD specific mutations. Moreover, the G175D mutant, selected previously for increased CYP2A6-C7H activity, also improved significantly CYP3A4-DBODF activities. The positive and pleiotropic effect of the G175D mutation was not observed with any of the other mutants but occurred for two of the substrates studied (coumarin and DBF). Therefore, binding of these substrates in different CYP isoforms may provoke similar structural alterations which then lead to better complex formation with the G175D FD mutant.

The inhibitory effect of cyt *c* on CYP dependent reactions confirmed the existence of coinciding conserved patches of acidic residues of CPR in its interaction with CYP and cyt *c*. The requirement for relatively high cyt *c*/CPR ratios to observe this effect indicates that the affinity of CPR for CYP is higher than that of CPR for cyt *c*. Our data also demonstrated that the amplitude of the cyt *c* effect is distinct for each CPR variant studied. Furthermore, the inhibition by cyt *c* is less effective with all CPR_mut_ previously reported to support a gain in activity for the CYP they were isolated when compared with CPR_wt_. Overall, our results confirm that the strengths of interactions between CYP isoforms and CPR are probably different, as suggested in our previous studies [11,16,18].

Structural effects of the four mutations were analyzed via molecular dynamics simulations. None of them affected drastically the stability or the secondary structures of the FD, in line with the facts that the effects of the mutations were quite specific and the effects measured rather small. Both P117H and N151D affect the electrostatic potential of the surface of the FD, albeit with opposite effects: While the histidine residue in P117H, by being involved in hydrogen bonding with either D116 or E117, slightly neutralizes the negatively charged surface, the presence of an aspartate in N151D, by adding a supplementary negative charge, increases the negative potential of the surface. These conserved acidic patches (D116, E118 and E119) were formerly suggested to play a role in CPR:CYP interactions [14,24,25,43,45,46], while the natural occurring variant A115V was shown to have CYP-isoform dependent effects as well [35,51]. N151D is also in the vicinity of the naturally occurring Q153R variant, a mutation linked with severely impaired steroidogenic activity, which has been shown to have CYP isoform dependent effects as well [33,35,51]. Molecular docking also demonstrated the existence of direct charge interactions between D116 and E118 of CPR with R440 of CYP2D6 [49,50]. Consequently, we hypothesize that the electrostatic potential around D116, E118, and E119 may control the formation of specific contacts between CYPs and CPR and thus, mutations that change the charge distribution modify the binding strength and impacts specifically some CYPs, but not necessarily the others.

In position 144 lays the highly conserved glycine position (100%, among 1221 CPR sequences [16]. This residue is in the vicinity of the conserved acidic patch D147/150, adjacent to the conserved tyrosine residue Y143 considered to be directly involved in FMN binding [26,52]. Additionally, G144 is located close to T142, a position found mutated in a natural occurring variant (T142A), reported to lead to a loss of activity in several CYPs, albeit in an isoform dependent manner [32,51]. Interestingly, while a cysteine residue at this position does not promote major structural rearrangements, the presence of an ionizable sulfur atom induces changes in the electrostatic potential around the FMN moiety. While we do not rule out that such changes also affect the redox potential of the FMN cofactor, the less negative electrostatic potential will also certainly modify the fine positioning of the FD into the CYP binding site prior to ET.

Finally, for the G175D mutant, the aspartate residue, being completely hindered within the structure, is almost non ionizable. However, the aspartate presence induces a small bending of β-strand 3, allowing for smaller φ angle values for A141 (but with larger standard deviations around the average value), impacting the fine positioning of the FMN cofactor.

## 4. Materials and Methods

### 4.1. Reagents

l-Arginine, thiamine, chloramphenicol, ampicillin, kanamycin sulfate, isopropyl β-D-thiogalactoside (IPTG) (dioxane-free), δ-aminolevulinic acid, cytochrome *c* (horse heart), glucose 6-phosphate, glucose 6-phosphate dehydrogenase, nicotinamide adenine dinucleotide phosphate (NADP+ and NADPH), resorufin, 7-hydroxy coumarin (7CH), 3-cyano-7-ethoxycoumarin (CEC), and fluorescein were obtained from Sigma-Aldrich (St. Louis, MO, USA). LB Broth, bacto tryptone and bacto peptone were purchased from BD Biosciences (San Jose, CA, USA). Bacto yeast extract was obtained from Formedium (Norwich, UK). Ethoxyresorufin and coumarin were obtained from BD Biosciences (San Jose, CA, USA) and dibenzylfluorescein from Santa Cruz Biotechnology (Santa Cruz, CA, USA). A polyclonal antibody from rabbit serum raised against recombinant human CPR obtained from Genetex (Irvine, CA, USA) was used for immune-detection of the membrane-bound CPR. All other chemicals and solvents were of the highest grade commercially available.

### 4.2. Membrane Fractions Preparation

The *Escherichia coli* cell model BTC was used for the heterologous co-expression of membrane-bound human CYP isoforms (CYP1A2, 2A6 or 3A4) together with human full-length CPR, using a biplasmid co-expression system, as previously reported [11,16,18,22,33,36,37]. CPR FD mutants P117H, G144C, G175D, and N151D, harbored in the pLCM_POR plasmid were used to transform *E. coli* PD301, already containing the expression vector (pCWori) for human CYP1A2, 2A6 or 3A4 (expressed in N-terminal modified forms), creating the following BTC strains: CPR_P117H_/CYP2A6, CPR_P117H_/CYP3A4, CPR_G144C_/CYP2A6, CPR_G144C_/CYP3A4, CPR_G175D_/CYP1A2, CPR_G175D_/CYP3A4, CPR_N151D_/CYP1A2, and CPR_N151D_/CYP2A6. Strains were cultured for heterologous expression of the human proteins [33,37,38], and membrane fractions of the different strains were prepared and characterized for total protein content (Bradford assay) as well as CYP- (CO-difference spectrophotometry) and CPR- (immune-detection by Western blotting) contents, as described previously [11,22,26,33,36,37,38].

### 4.3. CYP Enzyme Activity Assays

The catalytic activities of CYP1A2 (EROD), 2A6 (C7H), or 3A4 (DBODF), sustained by FD mutants were evaluated as reported previously, using 0–5 µM EthR (CYP1A2), 0–20 µM coumarin (CYP2A6), and 0–10 µM DBF (CYP3A4) [11,16]. When testing the effects of FD mutants in CYP1A2-activities (CECOD and DBODF), the concentrations ranges were 0–50 µM CEC and 0–37.5 µM DBF. Initial velocities were measured in triplicate and plots of velocity traces vs. substrates concentrations could be fitted according to the Michaelis–Menten equation (r^2^ ≥ 0.95) to determine steady-state kinetic parameters, using GraphPad Prism 5.01 Software (La Jolla, CA, USA) [22,33,38].

### 4.4. Cytochrome c Competition Assays

The effect of cyt *c* on CYP1A2, 2A6, or 3A4 activities, sustained by CPR_wt_ and CPR_mut_, was assessed using the same enzyme assay conditions as described above, except substrate concentrations were hold constant (5 µM EthR, 20 µM coumarin or 7.5 µM DBF), and a gradient of cyt *c* (0–40 µM for CYP1A2 and 3A4, 0–50 µM for CYP2A6) was applied in 100 mM potassium phosphate buffer (pH 7.6), and NADPH regenerating system (NADPH 200 µM, glucose 6-phosphate 500 µM and glucose 6-phosphate dehydrogenase 40 U L^−1^, final concentrations). Velocities were measured in triplicate in 96-well format using multi-mode microtiter plate reader (SpectraMax^®^i3x, Molecular Devices, USA; SoftMax Pro 2.0) [11,16]. All reaction velocities (*k*_obs_) were measured at least in triplicate [pmol of fluorescent product formed/(pmol of CYP per minute)].

### 4.5. Structural Analysis of CPR Mutants

The structural analysis of the four mutants (P117H, G144C, N151D and G175D) was performed using the CPR FD only, obtained from the crystal structure of human CPR_wt_ (PDB 5FA6) as the starting material. Mutations were constructed with the YASARA software (www.yasara.org). Fully hydrated molecular dynamics simulations were performed for 80 ns using a step size of 100 ps with the Yasara software suite using the AMBER14 force field at constant pressure (water density at 0.997 g/mL) and temperature (298 K). Parameters were set as follows: Cuboid cell extending 10 nm on each side of the protein, 0.9% NaCl and pH = 7.4. The Bio3D suite in R was used to calculate the coordinates of the average structure from all snapshots, the root mean square deviations (RMSD), the various reported distances and φ or ψ angles. Averages and standard deviations of the data was calculated with R. pKa of ionizable residues were calculated from the average structures using the AMBER force-field and the propka option of the pdb2pqr software at a pH of 7.5. Electrostatic potentials were calculated on the average structures with the integrated module of the Pymol software (www.pymol.org) using default parameters (non-linear Poisson-Boltzmann equation, protein dielectric 2.0, solvent dielectric 78.0, temperature 310 K, ion charge +1 of radius 2.0 Å and −1 of radius 1.8 both at 0.15 M) and plotted on the solvent excluded surfaces (Connolly surfaces). Figures were prepared with the Pymol software and R ggplot function. All modelling data is available upon request.

### 4.6. Statistical Analysis

Variance in data was analyzed through one-way ANOVA with Bonferroni’s multiple comparison test. The unpaired Student *t*-test was performed for calculation of the two-tailed *p* value. The analysis was performed with 95% confidence interval using the GraphPad Prism 5.01 Software. The significance level considered in all the statistical tests was 0.05.

## 5. Conclusions

Our data confirm and add additional evidences that CPR–CYP interactions are degenerated. While the purpose of this degeneracy is evident as only one donor serves multiple partners (and sometimes at the same time), it does not preclude the possibility of a gain of specificity, as demonstrated by the effects of our selected FD mutations. It is noteworthy that both surfaces (CPR and CYP) can be modified in order to get stronger interactions: Via mutations on the CPR side and via mutation and/or substrate binding on the CYP side. This latter effect may well explain how, in situation when specific CYP isoforms are metabolizing actively foreign or endogenous substrates, CPR could actually distinguish between non active and active CYPs, thus rendering CYP metabolism more efficient without any other transcriptional or posttranscriptional control, an elegant and efficient way to adjust CYP oxidative activities.

## Figures and Tables

**Figure 1 ijms-21-06669-f001:**
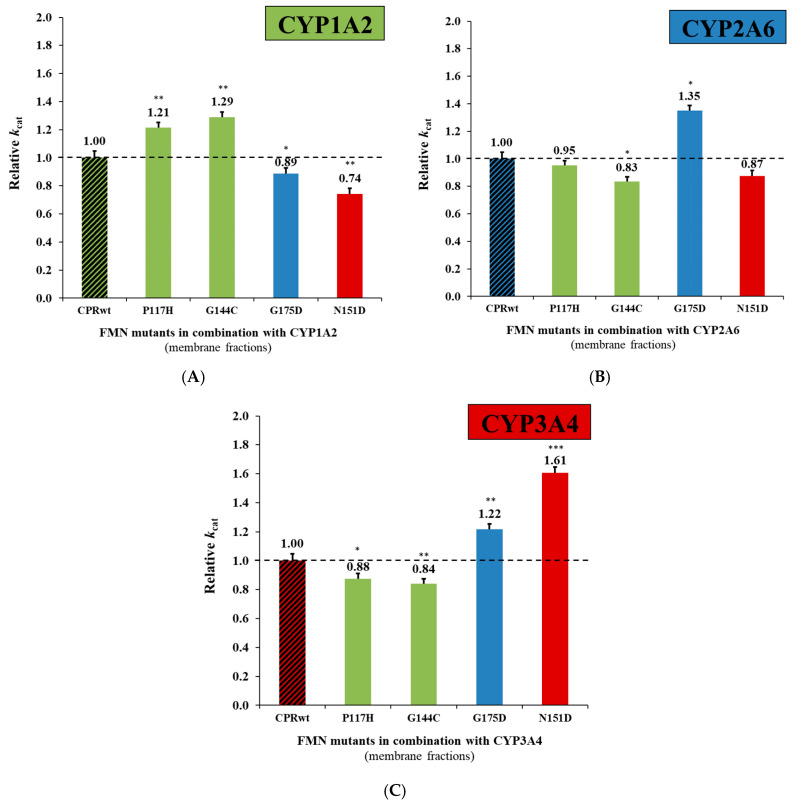
Relative turnover rates (*k*_cat_) (x fold) of the CPR_mut_:CYP1A2 EROD (**A**), CPR_mut_:CYP2A6 C7H (**B**), and CPR_mut_:CYP3A4 DBODF (**C**) activities, normalized by the *k*_cat_ demonstrated by CPR_wt_/CYP (determined in membrane fractions; technical replicates N = 3). CPR-FMN-domain wildtype (CPR_wt_:CYP, black stripes). CPR-FMN-domain mutants previously demonstrated to support a gain in CYP1A2-mediated EROD- (P117H and G144C: green), CYP2A6-mediated C7H- (G175D: blue), and CYP3A4-mediated DBODF-activity (N151D: red). *k*_cat_ values of the CPR_mut_/CYP were compared with the ones of the CPR_wt_/CYP applying the unpaired *t* test (*** *p* < 0.0005; ** *p* < 0.005; * *p* < 0.05).

**Figure 2 ijms-21-06669-f002:**
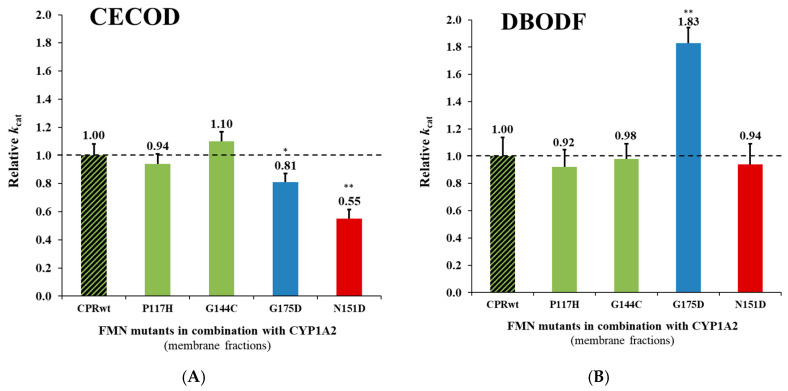
Relative turnover rates (*k*_cat_) (x fold) of the CPR_mut_:CYP1A2 CECOD (**A**), and DBODF (**B**) activities, normalized by the *k*_cat_ demonstrated by CPR_wt_/CYP (determined in membrane fractions) (technical replicates N = 3). CPR-FMN-domain wildtype (CPR_wt_:CYP, black stripes). CPR-FMN-domain mutants previously demonstrated to support a gain in CYP1A2-mediated EROD- (P117H and G144C, green), CYP2A6-mediated C7H- (G175D, blue), and CYP3A4-mediated DBODF-activity (N151D, red). *k*_cat_ values of the CPR_mut_/CYP1A2 were compared with the ones of the CPR_wt_/CYP1A2 applying the unpaired *t* test (** *p* < 0.005; * *p* < 0.05).

**Figure 3 ijms-21-06669-f003:**
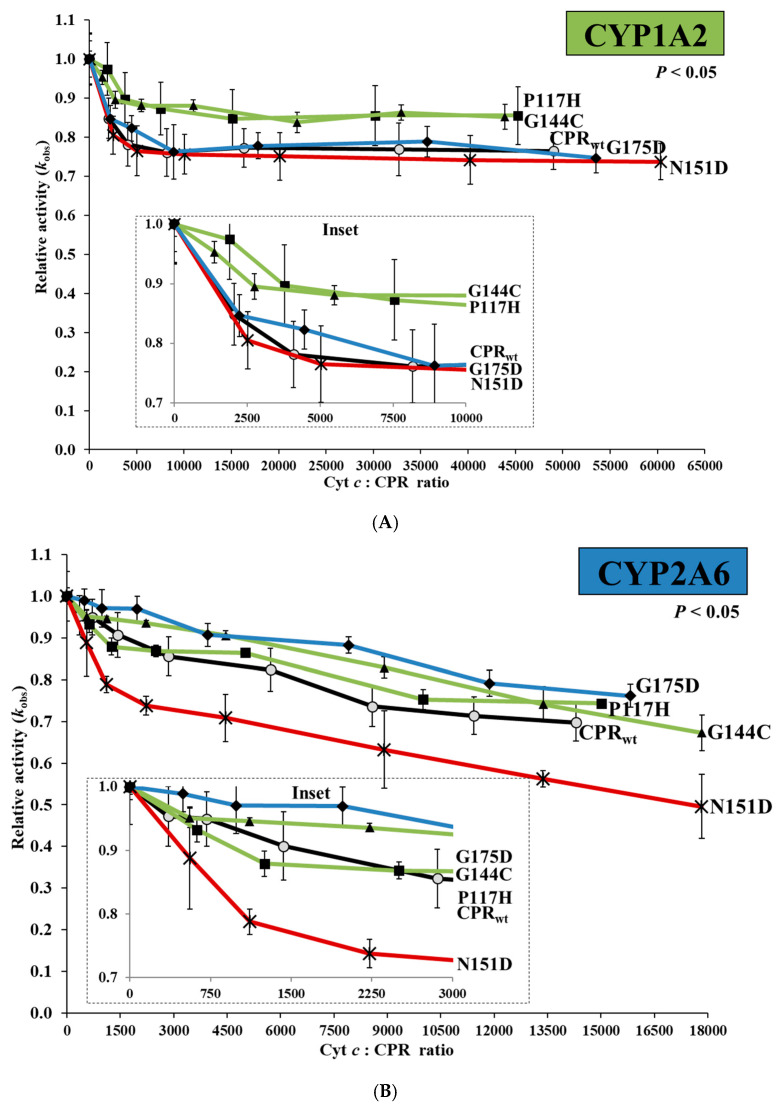
Relative CYP activities of the CPR-FMN-domain wildtype and four mutants plotted in function of the cyt *c*/CPR ratio. (**A**) CYP1A2 (EROD), (**B**) CYP2A6 (C7H), and (**C**) CYP3A4 (DBODF). Observed rate constants sustained by the CPR (*k*_obs_) were normalized by the CYP activity without cyt *c* [*k*_obs (0 nM cyt *c*)_]. FMN wildtype (CPR_wt_:CYP, black lines). FMN mutants previously demonstrated to support a gain in CYP1A2-mediated EROD- (P117H and G144C, green lines), CYP2A6-mediated C7H- (G175D, blue lines), and CYP3A4-mediated DBODF-activity (N151D, red lines). Normalized CYP activities represent the average of the three replicates and the error bars the standard deviation. Significant differences between the means of the activity profiles of CPR:CYP1A2 EROD (*p* = 0.015) (**A**) and CPR:2A6 C7H (*p* = 0.049) (**B**) were observed (ANOVA, one-way analysis of variance). (**D**) CPR_wt_ supporting CYP1A2-mediated EROD- (green line), CYP2A6-mediated C7H- (blue line), and CYP3A4-mediated DBODF-activity (red line).

**Figure 4 ijms-21-06669-f004:**
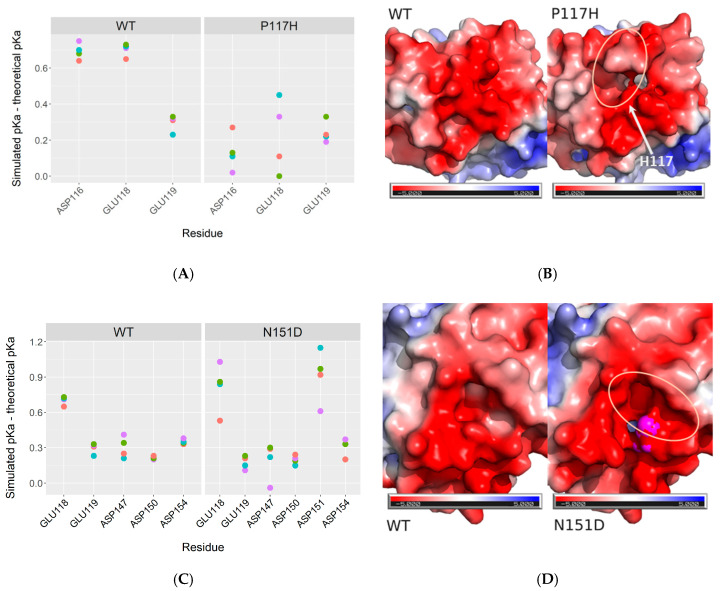
Electrostatics effects of the P117H and N151D mutations. Panels (**A**,**C**) present the variations of pKa of the surrounding aspartates and glutamates of P117H and N151D, respectively. pKas values were calculated on the average structures generated with four independent molecular dynamics simulations for wild-type (WT), P117H (P117H), or N151D (N151D) with the propka options of the pdb2pqr software, using a pH = 7.5 value and the AMBER force field. Theoretical pKa values of aspartate (3.8) or glutamate (4.5) were then subtracted from the simulated ones. Panels (**B**,**D**) present a mapping of the electrostatic potential onto the solvent excluded surface of P117H (Panel (**B**)) or N151D (Panel (**D**)) mutants compared to the wildtype FD. In Panel (**B**), the arrow represents the position of the H117 residue (hidden by the surface). In Panel (**D**), the magenta surface corresponds to the D151 residue. In Panels (**B**,**D**), the light orange oval shows the locations of the electrostatic potential and surface changes. Electrostatic potentials were calculated on the average structure of simulation 1 with the APBS module of Pymol using defaults parameters. The color scale is indicated below each surface and is in kT/e.

**Figure 5 ijms-21-06669-f005:**
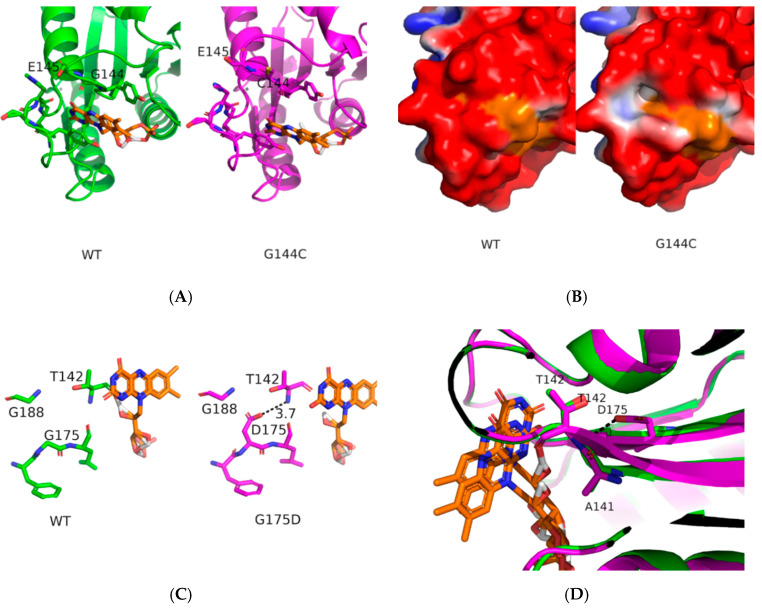
Effects of the G144C and G175D mutations. Panel (**A**) presents the wild-type (green, simulation 1) and G144C (magenta, simulation 1) average structures. Panel (**B**) presents a mapping of the electrostatic potential onto the solvent excluded surface of wild-type (left) or G144C mutant (right), the orange color corresponds to the surface of the flavin mononucleotide (FMN) residue. Panel (**C**,**D**) presents detailed view of the wild-type (green, simulation 1) and G175D (magenta, simulation 1) average structures.

**Table 1 ijms-21-06669-t001:** Cytochromes P450 (CYP) and cytochrome P450 oxidoreductase (CPR) contents of bacterial CPR/CYP co-expression system (BTC) membrane fractions.

Membrane Fractions		Protein Contents		
Cyp Isoform	CPR Form	CYP	CPR	CPR/CYP
		(pmol/mg Protein)		Ratios ^a^
**CYP1A2**	**wt ^a^**	54 ± 1	4.1 ± 1.5	1:13
	**P117H ^b^**	148 ± 1	12.2 ± 0.5	1:12
	**G144C ^b^**	124 ± 2	14.1 ± 0.2	1:9
	**G175D**	379 ± 2	26.5 ± 0.4	1:14
	**N151D**	446 ± 4	27.7 ± 0.5	1:16
	**CPR_null_^b^**	91 ± 6	NA	NA
**CYP2A6**	**wt ^b^**	139 ± 1	10.5 ± 1.3	1:13
	**P117H**	205 ± 3	16.5 ± 0.9	1:12
	**G144C**	315 ± 8	28.3 ± 0.8	1:11
	**G175D ^b^**	112 ± 6	11.3 ± 0.5	1:10
	**N151D**	221 ± 1	19.7 ± 0.8	1:11
	**CPR_null_^b^**	178 ± 4	NA	NA
**CYP3A4**	**wt ^a^**	83 ± 3	19.8 ± 0.2	1:4
	**P117H**	230 ± 9	35.5 ± 0.8	1:6
	**G144C**	189 ± 11	26.5 ± 0.3	1:7
	**G175D**	143 ± 5	31.5 ± 0.5	1:5
	**N151D ^b^**	45 ± 1	12.7 ± 0.6	1:4
	**CPR_null_^b^**	85 ± 3	NA	NA

CYP and CPR contents are mean ± sd (technical replicates N = 3). Color code indicates the CYP isoform with which the CPR-FMN-domain mutant was isolated (CYP1A2 green; CYP2A6: blue; CYP3A4: red). (NA) not applicable. ^a^ Co-expression of CPR_mut_ with CYPs was achieved with stoichiometry’s approximating those obtained with CPR_wt_. CPR/CYP ratios of the CPR_mut_ were not significantly different from the ones of the corresponding CPR_wt_ (*p* > 0.05), with each tested CYP. ^b^ Membrane fractions and values from batches used in our former studies [11,16].

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
