# Peer review of "Interaction Modes of Microsomal Cytochrome P450s with Its Reductase and the Role of Substrate Binding"

_ijms, 2020, doi:10.3390/ijms21186669_

Round 1

Reviewer 1 Report

Reviewer:
Recommendation: Major revision

Comments:
The manuscript by Esteves et al. presents mutational analysis of cytochrome P450 reductase (CPR) – FMN domain with three different cytochromes P450 (CYP1A2, CYP2A6, CYP3A4). The study combines the experiments with kinetics of typical reactions with structural analysis of mutations effect on CPR structure with use of molecular dynamics simulations. Experiments show possible modulation of CPR function by mutations that is connected by 4 different CPR mutations in combination with individual CYP isoforms. While the experiments show that the modulation of CYP reactions by CPR mutations is possible and can be specific, the structural analysis fails to mechanistically explain why. Similarly, the experiment show possible modulation of CYP1A2 reaction by different substrates, but again it fails to address the cause of such results.

As such, I think the results presented in the manuscript are interesting and timely, but I think the paper requires revision to address the following points before it can be considered for publication.

Major points:

Introduction:

  1. Page 2: List natural ratio CPR:CYP found in microsomal systems (as referenced on page 3, line 107)
  2. Page 2, line 77.: Interacting residues between CYP and CPR and charge interfaces were recently reviewed in https://doi.org/10.1016/j.jinorgbio.2018.03.002

Membrane fractions:

  1. Page 3, line 101: Authors claim that the most significant difference in CYP:CPR ratio can be found in CYP1A2:CPRG175D, however in Table 1. the more significant difference can be found in CYP1A2:CPRN151D.
  2. Page 3, Table 1. authors stated the stoichiometry ratios of CPR/CYP in membrane found after expression. Since those ratios vary among individual CYP isoforms, eg. 1:16 CYP1A2: CPRN151D and 1:4 CYP3A4:CPRN151D, it should be state how different CYP:CPR ratios effect the CYP activity and compare that ratio to natural one.

Catalytic activity:

  1. Page 4: In kinetic measurements authors concluded significantly lower activity for CPRG175CYP1A2 and equal activity for CPRN151DCYP2A6 with respect to CPRWT:CYP. This does not correspond to values presented in Figure 1A and Figure 1B: CPRG175CYP1A2: 89 – significantly lower activity CPRN151DCYP2A6: 0.87 – equal activity. Limits for protein activity measurements should be clearly set to avoid misleading conclusions. While it probably stems from the discussion of only new data reported in the article, it would be better to provide reader with the overview over ALL relevant results.
  2. No discussion is given for Michalis constants which also significantly differs according to Supplemental Tables S1 and S2.
  3. Page 6 - Figure 2B: different effect of G175D mutation with DBODF is not rationalized.

Cytochrome c competition:

  1. Cytochrome c competition with CPRmut/CYP chapter lacks some fundamental introduction. Is this the first study on cyt c/CPR competition? Why not use cytochrome b5 instead since this protein is involved in microsomal ET chain? Authors should comment on CPR/Cyt b5 competition.

Structural analysis of CPR mutants:

  1. Page 10, lines 235-240: Claim that some simulations may differ from original crystal structure is based on RMSD only. Since authors simulated the same system 4x 80 ns, it would be nice to show differences between individual final structures (possibly in SI) or rather to analyse clusters of structures coming from the simulations, since the average structure from simulation 1 is not entirely representative for whole dynamics coming from MD simulation (especially in P117H mutant where simulation 1 shows different trend with RMSD up to 3A to all other simulations with RMSD around 2A).
  2. Figure 4: It is not clear where exactly and how significantly the electrostatic potential between WT and mutants changes. Also, the position of mutation should be highlighted since current images do not show any visible differences upon mutations.
  3. Again, the different effect of mutations in respect to individual substrates should be explained by mechanistic explanation (possibly using docking) since the increase of reactivity is one of the most interesting results in the study.

General note:

  1. It would help FAIRness (Findable, Accessible, Interoperable, Reusable) of the publication of the data used within the study would be produced in a machine-readable format (CSV) with and models should be stored in public repository following the publication – such as ModelArchive.org and mutational databases.

Minor points:

  1. Page 3, line 106: explain shortcut BTC.
  2. Better graphical representation for Figure 3 would be appreciated. Maybe merging the inset in original graph. Also usage of different timelines for individual CYPs is not entirely helpful.
  3. Supplementary Figure S4 lacks units for RMSD.
  4. Page 10, line 248 – ND1 atom should be defined.
  5. Supplementary Figure S9 – increase the size of the font
  6. Molecular dynamics setup should be described more deeply in Materials and Methods, e.g. software used (GROMACS, AMBER?), NTV or NPT conditions etc. …

Nevertheless, the results are interesting, and after addressing the points raised above, the proposed work will be useful for the community interested in cytochrome P450 reactivity.

Author Response

Review's Comment: While the experiments show that the modulation of CYP reactions by CPR mutations is possible and can be specific, the structural analysis fails to mechanistically explain why. Similarly, the experiment show possible modulation of CYP1A2 reaction by different substrates, but again it fails to address the cause of such results.

Answer to comment: For mutations P117H, N151D, G144C we disagree with the reviewer’s comments. We did find a plausible explanation for the change in affinity with specific CYPs and this is explained in the text (pages 14-22, lines 247-365; new version). Basically all three mutations perturb the electrostatic potential in regions which were previously shown to affect CPR-CYP interactions, notably CYP2D6. We even discuss our results in the light of a docking simulation performed by the group of Ellis (Allorge D. et al. et al. Functional analysis of CYP2D6.31 variant: homology modeling suggests possible disruption of redox partner interaction by Arg440His substitution. Proteins 2005, 59, 339–346) that, upon docking CYP2D6 to CPR, described the salt bridge formation between D116 and E118 from the FMN domain with R440 from CYP2D6 (page 15, lines 287-290; new version). We do understand though that for the mutation G175D the strong biochemical effect does not fully translate to a strong structural modification, however, we still give hypotheses about the structural effects the mutation induces, notably on the FMN positioning and/or redox potential.

Reviewer's Comment: Similarly, the experiment show possible modulation of CYP1A2 reaction by different substrates, but again it fails to address the cause of such results.

Answer to comment: Our hypotheses are clearly stated in the discussion (pages 22-23, lines 384-393; new version): we wrote that “binding of these substrates in different CYP isoforms may provoke similar structural alterations which then lead to better complex formation with the G175D FD mutant.” It is now generally accepted that substrate binding in CYPs affect their proximal side, the location of interaction with redox partners. The most plausible explanation for the variation of G175D effects depending on the substrates is therefore linked to these possible structural variations induced by substrates binding. While we understand we cannot provide a better proof of this, we did address the potential cause.

Major points:

Introduction:

  1. Page 2: List natural ratio CPR:CYP found in microsomal systems (as referenced on page 3, line 107)

Answer (Q1): Physiological liver CYP:CPR ratios range from 1:2 to 1:13, indicating the submolar presence of CPR versus CYP, which is recapitulated in the BTC system. The authors would like to replace the first paragraph of the Results Section “2.1. Characterization of the membrane fractions” by “The characterization of contents of heterologous expressed proteins found in the membrane fractions (isolated using the bacterial CPR/CYP co-expression system, BTC) [22,36] containing the wildtype CPR (CPRwt) and mutant CPRs (CPRmut) combined with CYP1A2, 2A6 or 3A4 is shown in Table 1. Co-expression of CPRmut with CYP1A2, 2A6 or 3A4 was achieved with a stoichiometry approximating the one obtained with CPRwt for each CYP, demonstrating no significant differences (P > 0.05). As such, any difference detected in the CYP activities can be ascribed to differences in the CPRmut:CYP coupling. Indeed, the CPR:CYP stoichiometries measured in membrane fractions were similar to those described previously for the BTC system in our former studies [11,16,33,36–39], and comparable with the ranges of those reported for human liver microsomes, which range from 1:2 to 1:13 [40,41]. These ratios evidence a submolar presence of CPR versus CYP, which is recapitulated in the BTC system, as already demonstrated in our former studies [11,22,36,39].” (page 3, lines 97-111; new version).

  1. Page 2, line 77.: Interacting residues between CYP and CPR and charge interfaces were recently reviewed in https://doi.org/10.1016/j.jinorgbio.2018.03.002

Answer (Q2): The suggested reference has been added: (page 2, line 77; page 22, line 371; new version).

Membrane fractions:

  1. Page 3, line 101: Authors claim that the most significant difference in CYP:CPR ratio can be found in CYP1A2:CPRG175D, however in Table 1. the more significant difference can be found in CYP1A2:CPRN151D.

Answer (Q3): We would like to thank the reviewer for raising this point. Indeed, all CPRmut:CYP ratios were very similar to the ones obtained with CPRwt for each CYP, demonstrating no statistically significant differences (P > 0.05). We have the opinion that the small variation seen among CPR mutants and wildtype CPR with the same CYP isoform is insignificant (as now demonstrated). We previously demonstrated that, in these range, CPR:CYP ratios led to no significant differences in CYP-activity [11,16,33,36–38]. The authors would like to replace the first paragraph of the Results Section “2.1. Characterization of the membrane fractions” (page 3, lines 97-111; new version) (see our comment under Q1).

  1. Page 3, Table 1. authors stated the stoichiometry ratios of CPR/CYP in membrane found after expression. Since those ratios vary among individual CYP isoforms, eg. 1:16 CYP1A2: CPRN151D and 1:4 CYP3A4:CPRN151D, it should be state how different CYP:CPR ratios effect the CYP activity and compare that ratio to natural one.

Answer (Q4): As mentioned in our comments on Q1 and Q3, we have demonstrated that the observed CPR/CYP variations are i) insignificant and ii) observed variations of CPR/CYP have a negligible impact on CYP reaction velocities, as we have stated before. We have the impression that the Reviewer has interpreted all CPR/CYP ratios at the same time. However, the ratios of the CPR mutants should be compared with the ratio of the corresponding wildtype CPR/CYP (used as reference). Thus, CYP activity comparison was performed with CPRwt for the same CYP. I.e. CPR/CYP1A2 ratios of CPR mutants P117H, G144C, G175D and N151D with the one obtained for CRPwt/CYP1A2; CPR/CYP2A6 ratios of CPR mutants P117H, G144C, G175D and N151D with the one obtained for CRPwt/CYP2A6; and CPR/CYP3A4 ratios of CPR mutants P117H, G144C, G175D and N151D with the one obtained for CRPwt/CYP3A4.

Catalytic activity:

  1. Page 4: In kinetic measurements authors concluded significantly lower activity for CPRG175CYP1A2 and equal activity for CPRN151DCYP2A6 with respect to CPRWT:CYP. This does not correspond to values presented in Figure 1A and Figure 1B: CPRG175CYP1A2: 89 – significantly lower activity CPRN151DCYP2A6: 0.87 – equal activity. Limits for protein activity measurements should be clearly set to avoid misleading conclusions. While it probably stems from the discussion of only new data reported in the article, it would be better to provide reader with the overview over ALL relevant results.

Answer (Q5): Only statistically significant differences (P<0,05) were considered as limits for protein activity measurements comparison (CPRmutant/CYP vs corresponding CPRwt/CYP) (see our comment under Q4). Thus, statistically significant lower activity for CPRG175/CYP1A2 was observed, when compared with CPRWT/CYP1A2 (P < 0.05), and no statistically significant differences were observed for the CPRN151D/CYP2A6 activity, when compared with CPRWT/CYP2A6 (P > 0.05).

  1. No discussion is given for Michalis constants which also significantly differs according to Supplemental Tables S1 and S2.

Answer (Q6): We would like to thank the reviewer for raising this point. The authors would like to add the sentence “In few cases, the apparent affinity constant (KM) changed with the FD mutant CPRs. The apparent affinity constant of the CYP enzyme complex is a combination of multiple factors such as substrate entrance, binding and product egress but also ET rates. We thus hypothesize that mutations in the FD, by either affecting the binding of CPR to CYPs or directly modifying ET, may affect the apparent affinity constants.” in the Results subsection “2.2.1. Reaction velocities of the different CPRmut:CYP combinations” (page 4, lines 133-138; new version). However, it is the authors opinion that kcat is the most indicative kinetic parameter to study changes in the interaction between CPR and CYP because KM, as written in the text, is a combination of multiple factors (affinity and ET changes) where kcat is mainly dependent on the ET efficiency.

  1. Page 6 - Figure 2B: different effect of G175D mutation with DBODF is not rationalized.

Answer (Q7): The different effect of the G175D mutation on the DBODF activity versus the other CPR mutants is indeed surprising but can only properly rationalized with substrate docking modelling experiments, which would be the subject of a whole new study.

Cytochrome c competition:

  1. Cytochrome c competition with CPRmut/CYP chapter lacks some fundamental introduction. Is this the first study on cyt c/CPR competition? Why not use cytochrome b5 instead since this protein is involved in microsomal ET chain? Authors should comment on CPR/Cyt b5 competition.

Answer (Q8): The competing capacity of CYB5 with CPR for binding to CYP has been used to verify increase affinity of the CPR mutants, as demonstrated in our recent reports [11,16]. However, the CYB5 competition data is rather complex, and difficult to interpret. CYB5 has two effects on CYP activity, of which one is confounding, namely an initial stimulating effect (depending on the CYP isoform), as demonstrated in our former publication [16]. Several CPR-FMN-domain residues are involved in the interaction with both redox partners (cyt c and CYP). Differences in affinity of the CPR-FMN-domain mutants in binding with CYP may therefore be studied by cyt c competition experiments, based on that these redox partners will mutually compete for, at least partially, overlapping CPR binding sites. Therefore, cyt c titration has a pure competitive inhibition effect on CYP mediated reactions as it directly competes with CYP for CPR electrons, as evidenced by our results. A new introductory part for subsection ”2.3. Cytochrome c competition with CPRmut/CYP” has now been introduced in the manuscript “In two recent studies, we applied cytochrome b5 (CYB5) to probe affinity differences between CPR mutants and CYP [11, 16]. Although informative, the inhibition data were difficult to interpret due to a dual effect by CYB5, namely an initial stimulating effect on CYP activity (depending on the CYP-isoform) and subsequent inhibitory effects at higher CYB5 levels when CPR binding is prevented [16]. An alternative competitive inhibitor is cyt c.” and “Therefore, cyt c competition experiments may shed light on differences in affinity between the various FD mutants when binding to different CYPs, without any confounding stimulatory effects when CYB5 is applied for this purpose.” (pages 7 and 8, lines 191-195 and 198-200; new version).

Structural analysis of CPR mutants:

  1. Pages 14-15, lines 247-253: Claim that some simulations may differ from original crystal structure is based on RMSD only. Since authors simulated the same system 4x 80 ns, it would be nice to show differences between individual final structures (possibly in SI) or rather to analyse clusters of structures coming from the simulations, since the average structure from simulation 1 is not entirely representative for whole dynamics coming from MD simulation (especially in P117H mutant where simulation 1 shows different trend with RMSD up to 3A to all other simulations with RMSD around 2A).

Answer (Q9):

Page 10, lines 235-240: Claim that some simulations may differ from original crystal structure is based on RMSD only.

The claim that some simulations differ from crystal structure is not really the message we delivered : “A first analysis of root mean square deviations (RMSD) along the progress of the simulations revealed a common pattern: a rise in the RMSD in the first 20 ns, corresponding to an increase in the energy as the starting coordinates were directly issued from a crystal structure and then a stabilization of the RMSD corresponding to a stable energy state, with rather small variations between the different simulations” (page 15, lines 249-253; new version). The fact that the RMSD in the beginning of the simulations rise is normal when starting from a crystal structure which represents a “frozen state”. We even wrote: “Therefore, the various selected mutations do not, in an 80 ns simulation, perturb the overall structure or folding of the FD, but probably have rather specific and localized effects” (page 15, lines 264-266; new version) which we demonstrate afterwards.

Comment: Since authors simulated the same system 4x 80 ns, it would be nice to show differences between individual final structures (possibly in SI) or rather to analyse clusters of structures coming from the simulations, since the average structure from simulation 1 is not entirely representative for whole dynamics coming from MD simulation (especially in P117H mutant where simulation 1 shows different trend with RMSD up to 3A to all other simulations with RMSD around 2A).

The use of the final structure in an 80 ns simulation is not adequate. First, in all simulations, we reached equilibrium (as demonstrated with the stabilization of the RMSD). The last snapshot would then only be a representative of one structure amongst many others rather than the ensemble. The use of the average structure is therefore better. We actually performed cluster analyses of the simulations but could not find visible differences between the first three clusters as determined by a principal component analysis with the Bio3D suite package. The fact that the RMSD jumps to different values at the beginning of the simulation shows that the initial conditions induced by the simulations differ between one to another, a normal feature in independent simulations. However, it is true that the Simulation 1 of P117H is quite different from all others and this is probably due to the initial conditions at the start of the dynamics simulation. We compared, pairwise, all average or minimum energy structures and calculated all RMSD values. These RMSD values range from 0.27 to 2.39 Å (datafile available on request). The maximal values are found with P117H simulation which is quite logical but for all other structure comparisons, RMSD values are below 2 Å. We added a figure in the SI to show that average structures that were used are quite similar to the starting crystal structure of the human FMN domain of CPR, at least on a global analysis. Similar and even higher RMSD values were found for the FAD domain alone by the group of Oostenbrink (Sündermann, A.; Oostenbrink, C. Molecular dynamics simulations give insight into the conformational change, complex formation, and electron transfer pathway for cytochrome P450 reductase. Protein Sci. Publ. Protein Soc. 2013, 22, 1183–1195) in their simulation of the entire CPR in different states. As suggested by the reviewer, a supplementary Figure comparing all average simulations with the FMN domain from the crystal structure was added (Figure S5). Modifications in the text were also added to account for the reviewer’s suggestions (page 15, lines 254-264; new version).

  1. Figure 4: It is not clear where exactly and how significantly the electrostatic potential between WT and mutants changes. Also, the position of mutation should be highlighted since current images do not show any visible differences upon mutations.

Answer (Q10): Figure 4 was adapted, zoomed and modified according to wishes of the reviewer.

  1. Again, the different effect of mutations in respect to individual substrates should be explained by mechanistic explanation (possibly using docking) since the increase of reactivity is one of the most interesting results in the study.

Answer (Q11): We wrote a tentative explanation of this fact in the discussion: “This implies that the structural determinants present in the proximal side of CYPs can be modulated by the type of substrate, which, in return, may change CPR:CYP interactions, potentially annihilating the specific effects of the FD specific mutations. Moreover, the G175D mutant, selected previously for increased CYP2A6-C7H activity, also improved significantly CYP3A4-DBODF activities. The positive and pleiotropic effect of the G175D mutation was not observed with any of the other mutants but occurred for two of the substrates studied (coumarin and DBF). Therefore, binding of these substrates in different CYP isoforms may provoke similar structural alterations which then lead to better complex formation with the G175D FD mutant.” (pages 22-23, lines 386-393; new version). The fact that substrates can modulate the strength of interaction between CPR and CYP is related to the subtle structural changes induced by substrate binding on CYPs. We understand the comment of the referee and also think a docking analysis could help better understand this phenomenon. However, we believe this would be the subject of another paper as such docking analyses are extremely dense and time consuming.

General note:

  1. It would help FAIRness (Findable, Accessible, Interoperable, Reusable) of the publication of the data used within the study would be produced in a machine-readable format (CSV) with and models should be stored in public repository following the publication – such as ModelArchive.org and mutational databases.

Answer (Q12): ModelArchive is a repository that is mostly used to put modelled structures of proteins for which no structural data is available. In this sense, the structural information of the mutants of the FMN domain is not totally adequate for this repository since: i) they only represent a part of the CPR, ii) they only contain one modification and iii), the most important information comes from the simulation, not from any snapshot of a single structure. For mutational database, to our knowledge, the Protein Mutant Database does not exist anymore. However, all modelling data are freely available and can be requested from the authors at any time (see Materials and Methods, subsection “4.5. Structural analysis of CPR mutants”, page 25, lines 497-498; new version).

Minor points:

  1. Page 3, line 106: explain shortcut BTC.

Answer: The text “(isolated using the bacterial CPR/CYP co-expression system, BTC) [22,35]” was added to the first paragraph of the Results section “2.1. Characterization of the membrane fractions” (page 3, line 98; new version).

  1. Better graphical representation for Figure 3 would be appreciated. Maybe merging the inset in original graph. Also usage of different timelines for individual CYPs is not entirely helpful.

Answer: Figure 3 has been adapted. We have the impression that the Reviewer has interpreted the x axis unit as time. The relative activity of the CPR/CYP couples was plotted against the cyt c:CPR ratio. The different ratios plotted on the X-axis are due to difference in sensitivity among the individual CYP isoforms for cyt c inhibition.

  1. Supplementary Figure S4 lacks units for RMSD.

Answer: Figure S4 has been adapted and the unit of RMSD is now printed.

  1. Page 10, line 248 – ND1 atom should be defined.

Answer: although the ND1 nomenclature is generally accepted by people working on protein structure, the text was changed considering the remark (page 15, line 272; new version).

  1. Supplementary Figure S9 – increase the size of the font

Answer: Figure S9 has been adapted.

  1. Molecular dynamics setup should be described more deeply in Materials and Methods, e.g. software used (GROMACS, AMBER?), NTV or NPT conditions etc. …

Answer: The molecular simulations are now described with more details (see Materials and Methods, subsection “4.5. Structural analysis of CPR mutants”), “Fully hydrated molecular dynamics simulations were performed for 80 ns using a step size of 100 ps with the Yasara software suite using the AMBER14 force field at constant pressure (water density at 0.997 g/ml) and temperature (298 K). Parameters were set as follows: cuboid cell extending 10 nm on each side of the protein, 0.9% NaCl and pH = 7.4.” (page 24, lines 485-488; new version).

Reviewer 2 Report

The paper remarkably adds to the problem of degenaracy of protein-protein interaction, in the case of CPR-CYP interactions. Experiments indicate via mutagenesis in the active site enviroment which mutations do or do not affect electron transfer and docking experiments confirm results.

I do have only two  major questions taht possibly where already addressed by the authors in previous publications on the same topic:

a) to which extent the expression system may or may not affect protein conformations

b) as to the docking procedure, did the authors consider to use other methods? In other words would results be confirmed also by other computational approaches?

Author Response

Reviewer's comment: I do have only two  major questions taht possibly where already addressed by the authors in previous publications on the same topic:

a) to which extent the expression system may or may not affect protein conformations

Answer (Qa): We would like to thank the reviewer for raising this point. Human CPR/CYP couples were co-expressed in a bi-plasmid system using the BTC model, as described in the subsection “4.2. Membrane fractions preparation” in Materials and Methods (page 24, lines 447-449; new version). The usefulness of the bacterial BTC model for the co-expression of human CPR and CYPs have been demonstrated in various former reports, which indicated CYP enzyme kinetics equal or very similar to those obtained with liver microsomes. These results implies that this expression system has none or at least no significant effect on the conformation of the expressed proteins, which would result in a different kinetic behavior. This is probably best demonstrated in our previous report by Moutinho et al., 2012 [33].

b) as to the docking procedure, did the authors consider to use other methods? In other words would results be confirmed also by other computational approaches?

Answer (Qb): Molecular dynamics simulation is a general and accepted method to decipher the structural variations of mutations. We did not perform any docking experiment. This type of experiments would be the next logical step. However, we believe such kind of experiments would be fit for another study as these approaches are extremely dense and time-consuming.

Round 2

Reviewer 1 Report

Authors responded to all of my previous concerns and dissolved them. I have no further concerns. 

Karel Berka